# Antitumor Potential and Phytochemical Profile of Plants from Sardinia (Italy), a Hotspot for Biodiversity in the Mediterranean Basin

**DOI:** 10.3390/plants9010026

**Published:** 2019-12-24

**Authors:** Concettina Cappadone, Manuela Mandrone, Ilaria Chiocchio, Cinzia Sanna, Emil Malucelli, Vincenza Bassi, Giovanna Picone, Ferruccio Poli

**Affiliations:** 1Department of Pharmacy and Biotechnology, University of Bologna, via San Donato 19/2, 40127 Bologna, Italy; concettina.cappadone@unibo.it (C.C.); emil.malucelli@unibo.it (E.M.); vincenza.bassi@studio.unibo.it (V.B.); giovanna.picone2@unibo.it (G.P.); 2Department of Pharmacy and Biotechnology, University of Bologna, Via Irnerio, 42, 40126 Bologna, Italy; ilaria.chiocchio2@unibo.it (I.C.); ferruccio.poli@unibo.it (F.P.); 3Department of Life and Environmental Sciences, University of Cagliari, Via Sant’Ignazio da Laconi 13, 09123 Cagliari, Italy; cinziasanna@unica.it

**Keywords:** Sardinian plants, antitumor activity, cell cycle, apoptosis, NMR profiling, sesquiterpene lactones

## Abstract

Sardinia (Italy), with its wide range of habitats and high degree of endemism, is an important area for plant-based drug discovery studies. In this work, the antitumor activity of 35 samples from Sardinian plants was evaluated on human osteosarcoma cells U2OS. The results showed that five plants were strongly antiproliferative: *Arbutus unedo* (AuL), *Cynara cardunculus* (CyaA), *Centaurea calcitrapa* (CcA), *Smilax aspera* (SaA), and *Tanacetum audibertii* (TaA), the latter endemic to Sardinia and Corsica. Thus, their ability to induce cell cycle arrest and apoptosis was tested. All extracts determined cell cycle block in G2/M phase. Nevertheless, the p53 expression levels were increased only by TaA. The effector caspases were activated mainly by CycA, TaA, and CcA, while AuL and SaA did not induce apoptosis. The antiproliferative effects were also tested on human umbilical vein endothelial cells (HUVEC). Except for AuL, all the extracts were able to reduce significantly cell population, suggesting a potential antiangiogenic activity. The phytochemical composition was first explored by ^1^H NMR profiling, followed by further purifications to confirm the structure of the most abundant metabolites, such as phenolic compounds and sesquiterpene lactones, which might play a role in the measured bioactivity.

## 1. Introduction

Natural products (NP)-based drug discovery strongly benefits from research conducted in regions with high biodiversity and endemism [1,2]. In this context, Sardinia (Italy) deserves particular attention. This island is a hotspot for biodiversity, with a wide range of habitats and high degree of endemism, due to its geographical isolation and high geological and geomorphological diversification [3,4]. 

Sardinian flora consists of 2441 taxa [5], of which 295 taxa are endemic [6]. Among them, 189 are exclusive to Sardinia, 90 are Sardinian–Corsican endemics, and 16 taxa are also present in the Tuscan Archipelago [6]. Even though scientific evidence confirmed their interesting phytochemical and biological features [7,8,9,10], to date most Sardinian endemic plants remained scantly or not at all investigated.

In addition, plant-based traditional medicine is still widely practiced and documented in Sardinia (Table 1). Ethnobotany plays also an important role in NP-based drug discovery, providing precious information about plant properties and uses, thus, increasing the chances to individuate active natural products with a good safety profile. 

Several studies have demonstrated the role of medicinal plants in prevention and treatment of cancer [11], one of the leading causes of morbidity and mortality worldwide, responsible for one in eight deaths worldwide [12]. Chemotherapy and radiotherapy, routinely used for cancer treatment, are not devoid of their own intrinsic problems, such as the scarce selectivity toward cancer cells or the onset of drug resistance, requiring further research and treatment development. Next to common targets for tumor therapy, such as cell cycle inhibitors or proapoptotic agents, there are often angiogenesis modulators, especially in combined treatments [13]. The solid tumors, in fact, require an increased blood supply to support their growth, thus, angiogenesis is critical for the initiation, growth, and metastasis of these tumors [14].

As previously mentioned, in cancer treatment, the contribution of natural drugs has been both historically and currently remarkable [11,15]. Moreover, the preventive or antitumor activity of plant extracts was explored, attributing the result to the combined action of various phytochemicals rather than a single molecule [16,17]. Epidemiological studies also established associations between certain dietary patterns and reduced cancer risk [18,19], and significant results were obtained in vitro and in vivo on different food secondary metabolites such as carotenoids, phenolic, and organosulfur compounds [20,21]. 

On this basis, this work aimed at exploring the in vitro anticancer potential of a wide number of Sardinian plants, investigating the mechanisms of activity and the phytochemical profiles of the most active ones.

## 2. Results and Discussion

### 2.1. Plant Traditional Uses and Screening of Antiproliferative Effect 

Firstly, a literature survey on uses in traditional Sardinian medicine of all the analyzed plants was conducted. The results are reported in Table 1. 

Out of 30 plants, 18 resulted in being widely and commonly used for medicinal or nutritional purposes, while 13 are endemic and little known. For five of them (*Centaurea horrida*, *Ferula arrigonii*, *Hypericum scruglii, Limonium morisianum*, and *Plagius flosculosus*), there are no literature data available on their uses in traditional medicine, probably because of their rarity. They were included in the study for their importance as endemic species of Sardinia. 

As a first line of screening, all plant extracts underwent MTT, 3-(4,5-dimethylthiazol-2-yl)-2,5-diphenyltetrazolium bromide assay to evaluate their growth inhibition activity on U2OS cells. Cells were treated for 24 h with plant extracts at two fixed concentrations (50 and 100 µg/mL), and the extracts which reduced cell growth by at least 20% were considered promisingly active and selected for further investigations. On this basis, five extracts significantly reduced osteosarcoma cell viability, namely: *Arbutus unedo* (AuL), *Centaurea calcitrapa* (CcA), *Cynara cardunculus* (CycA), *Smilax aspera* (SaA), *Tanacetum audibertii* (TaA) (Figure 1A).

To calculate the EC_50_ concentration of the selected plants, dose–response curves at different times of treatment were performed (Appendix A). The best-fitting sigmoidal function was obtained at 48 h of treatment (Figure 1B). Hence, all following experiments were carried out treating the cells for 48 h with the extracts at their EC_50_ concentration.

### 2.2. Effect of the Selected Plants on the Cell Cycle Progression

In order to investigate the mechanism of antiproliferative effects induced by the most active plant extracts, the cell cycle analysis was performed. Treated and control cells were stained with propidium iodide and analyzed by flow cytometry. All five extracts determined a block of cell population in G2/M phase, even though of different entities, and as a consequence a variation of cell percentage in G0/G1 and S phases. In particular, the increase of cell percentage in G2/M phase became more consistent after treatment with CycA and TaA, reaching values above 50% (Figure 2A). 

Several anticancer agents induce cell cycle arrest involving the pathway of the tumor suppressor p53 [34]; thus, the expression levels of this protein were analyzed by immunofluorescence after treatment with CycA and TaA. The TaA extract significantly affected p53 levels: the fluorescence mean channel went from 18 to 30 on a logarithmic scale, corresponding to an increase of the protein levels by 60% compared to the controls. On the contrary, CycA did not involve the p53 protein, as its levels in the treated cells were comparable to those of the controls (Figure 2B). Therefore, the two extracts exerted their effects through different mechanisms, involving various targets and signal transduction pathways. It is well known that many proapoptotic agents, including natural compounds and chemotherapeutics drugs, are able to induce apoptosis without any changes in p53 levels [35,36]. In particular, it is reported that doxorubicin-induced apoptosis on p53-null human osteosarcoma cells was characterized by an increase of ROS production, suggesting that ROS might act as the signal molecules even in the absence of p53 upstream cell death process [37]. 

### 2.3. Assessment of Apoptotic Effect of Selected Extracts 

To give some insights into the biological effects of the active extracts, it was determined whether they induced apoptosis. Firstly, the morphology of the treated cells was evaluated. As expected, a significant reduction of adherent cells was visible as well as many floating cells, especially with CycA and TaA extracts (Figure 3A). Cells were stained with HOECHST nuclear probe to discriminate between apoptotic and necrotic death: the typical nuclear fragmentation of apoptotic cells, associated with the higher fluorescence intensity, was evident in treated samples, confirming the more prominent cytotoxicity of the CycA and TaA extracts (Figure 3B).

It is well known that activation of effector caspases represents a clear marker of apoptotic cell death [38]. Hence, to evaluate the apoptotic effect of the selected extracts on osteosarcoma cells, the caspase-3 activity was determined. The histogram in Figure 3C shows that three plants triggered a significant caspase activation, although with different degrees. However, the higher induction was observed with CycA, followed by TaA and finally by CcA extracts. On the contrary, the antiproliferative effects of AuL and SaA extracts did not culminate in apoptotic events and these latter can be considered cytostatic rather than cytotoxic agents.

### 2.4. Comparison of the Selected Plants’ Effects on U2OS and HUVEC Cells

A growing number of research has shown that angiogenesis is a hallmark of tumor development, becoming an attractive target for anticancer chemotherapy [39,40]. Furthermore, inhibiting angiogenesis before it starts (angio-prevention) allows blocking the expansion of hyperplastic foci and subsequent tumor development at the premalignant stage. Endothelial cell proliferation and migration are the key steps of the angiogenic process. Therefore, we investigated the effects of the most active extracts on the proliferation of human endothelial vein cells. As previously stated for Figure 1, a reduction of cell viability of at least 20% was considered significant. The data obtained clearly showed that four out of five tested extracts induced a marked cytotoxicity also in endothelial cells. In particular, CcA and CycA reduced the cell viability by 90% at both 50 μg/mL and 100 μg/mL concentrations. SaA and TaA reduced the viability by 30% and 50%, respectively, only at 100 μg/mL concentration, similarly to osteosarcoma U2OS cells. Although this preliminary result requires further investigations, our findings suggest additional activity of these extracts, useful to contrast tumor growth. On the other hand, the lack of toxicity of AuL against HUVEC cells makes this extract interesting for the selectivity toward tumor cells (Table 2).

### 2.5. Phytochemical Analyses 

Since the results showed that *Arbutus unedo* (leaves) (AuL), *Cynara cardunculus* (aerial parts) (CycA), *Centaurea calcitrapa* (aerial parts) (CcA), *Smilax aspera* (aerial parts) (SaA), and *Tanacetum audibertii* (aerial parts) (TaA) were promising for their biological activities, they were subjected to further phytochemical investigations. Three of them (CycA, CcA, TaA) belong to the Asteraceae family and, among them, *T. audibertii* is endemic to Sardinia and Corsica. 

Firstly, ^1^H NMR profiling was performed to acquire a preliminary overview of the main metabolites present in the extracts (Figure 4). Both secondary and primary metabolites (such as amino acids, carbohydrates, and organic acids) were detected in the extracts (diagnostic ^1^H NMR signals are listed in Appendix A). The content of the main metabolites was then determined, on the basis of the internal standard TMSP, as reported in Table 3.

Regarding secondary metabolites, AuL showed a high concentration of arbutin, a glycoside hydroquinone which is considered the main active principle of this plant [41]. The NMR profiling highlighted also the presence of *O*-rhamnosyl flavonoid, showing typical signals of α-rhamnose, namely the doublet of the methyl group resonating around δ 0.89. Both arbutin and flavonoids might play a role in the measured biological activities [42,43].

Since, in the first screening, both leaves (AuL) and fruits (AuF) of *Arbutus unedo* were tested, and only leaves were found active, the ^1^H NMR-based phytochemical profiles of these two samples were also compared. The profile of AuF lacked both arbutin and rhamnosyl flavonoid signals (Appendix A), supporting the importance of these compounds for the bioactivity of AuL. 

From the ^1^H NMR profile of CycA, the guaiane-type sesquiterpene lactone cynaropicrin resulted in being the most abundant metabolite (see Table A1 and Appendix A for structure elucidation). 

Interestingly, despite its high lipophilicity, this compound was found highly concentrated (159.4 μg/mg of extract) in the extract solubilized in D_2_O-buffer. This phenomenon is quite common in natural products chemistry, since the inclusion of a lipophilic compound in a complex mixture of metabolites, such as a plant extract, often increases its solubility in water [44]. The discrete concentration of cynaropicrin found in the water medium is interesting also because CycA is traditionally prepared in the form of a decoction or infusion (Table 1).

Cynaropicrin is potentially one of the most important active principles responsible for the bioactivity of CycA [45]. Similarly to CycA, the bioactivity of the other plants belonging to Asteraceae might be due to sesquiterpene lactones, which are typical of this plant family, and are reported to possess antitumor potential [46]. Moreover, several studies report the ability of these plant extracts to inhibit the proliferation mainly of breast cancer cells, but also of hematological cancer [47,48]. 

Regarding CcA, the ^1^H NMR profile clearly presented signals ascribable to sesquiterpene lactones [48]. In particular, the geminal protons of the double bond in position 13 on the γ-lactone were clearly visible in the CcA profile. They showed two doublets with chemical shift of δ 6.05 and 5.59 and coupling constant of 3.23 Hz (calculated by J-res experiment), with HSQC correlation to the carbon at δ 119.24, and HMBC and COSY correlation to carbon at δ 52.72 and proton at δ 3.08, respectively, which are characteristic chemical shifts of position 7 of this nucleus. The latter carbon, in turn, showed HMBC correlation to a proton resonating at δ 2.50 (linked to carbon at δ 51.41), which is typical of position 5 of the molecule (Appendix A). High antioxidant and cytotoxicity activities on HeLa and Vero cell lines were reported for *Centaurea calcitrapa* [49]. Our results confirm the capability of the hydroalcoholic plant extract to inhibit cell proliferation on another cell type, highly proliferating and undifferentiated such as the U2OS cell line.

Interestingly, in this work, two other species of *Centaurea* were tested (CnA and ChA) showing no relevant bioactivity. The ^1^H NMR profiles of all three *Centaurea* species were recorded and compared. In the ChA profile, no signals ascribable to γ-lactone were visible, while in CnA, they were present, showing a lower intensity compared to the CcA profile (Appendix A). This data might support the importance of these compounds for the measured bioactivity. 

The phytochemical profile of SaA showed a high quantity of shikimic acid and quinic acid, together with caffeic acid. There are very few studies on potential antiproliferative of *Smilax aspera*. It is reported that steroidal saponins and anthocyanins have been isolated from this plant, and that some of these compounds exhibit cytotoxic activity against human normal amniotic and human lung carcinoma cell lines [50,51]. We demonstrated that it induces cytostatic effects on osteosarcoma cells.

Moreover, flavonoid and phenolic content of the five selected plants were determined, and the results were compared by one-way ANOVA test (Appendix A). The phenolic content ranged from 76.97 to 25.68 mg GAE/g of extract, with the following order of concentration: AuL > SaA > TaA > CycA > CcA. Total flavonoid content ranged from 52.3 to 16.09 mg RE/g of extract. No significant differences were found among the samples, with the exception of TaA, which resulted as the extract with the highest content of flavonoids. It is noteworthy that samples treated with this latter extract showed a marked apoptosis induction, characterized by significant G2/M arrest and increased p53 level. The high phenolic content could explain the clear proapoptotic effect, as it is well known that this class of compounds is able to induce apoptosis via the p53 pathway [39]. Our work demonstrated for the first time the antitumor potential of *Tanacetum audibertii*, as only antifungal properties have been documented in the literature [52].

## 3. Materials and Methods 

### 3.1. Chemicals

All reagents were purchased from Sigma Aldrich (Milano, Italy), except the deuterated solvents, which were purchased by Eurisotop (Cambridge, UK). 

### 3.2. Plant Material

All plants were collected at the flowering stage. Species were botanically identified by Dr. Cinzia Sanna and voucher specimens were deposited at the General Herbarium of the Department of Life and Environmental Sciences, University of Cagliari (Table 4). Regarding selected endemic species, they are not protected by local or international regulations. Furthermore, the locations where they were harvested are not included in national or local parks or any other natural protected areas. Therefore, no specific permission was required for their collection.

### 3.3. Preparation of Plant Extracts for Bioactivity Tests and for ^1^H NMR Profiling 

A total of 30 mg of dried and powdered plant material were extracted by sonication for 30 min using 1.5 mL of MeOH/H_2_O (1:1). Subsequently, samples were centrifuged (1700× *g*) for 20 min and the supernatant was separated from the pellet and dried, firstly in vacuum concentrators (speedVac SPD 101b 230, Savant, Italy) for 2 h to remove MeOH, then the residual extracts were freeze-dried overnight to completely remove the residual H_2_O, finally yielding the crude extracts. This extraction procedure is ideal to prepare a small quantity of extracts for in vitro bioactivity tests, thus for screenings of a high number of plants, allowing a minimal consumption of both solvents and plant material. The choice of the extraction solvents was based on metabolomics works [53,54], where MeOH/H_2_O (1:1) turned out as the best choice for first-line extraction of generic plant material, having obtained a broad spectrum of compounds.

Extracts were solubilized in H_2_O to prepare stock solution at a concentration of 1 mg/mL, which was centrifuged for 10 min (1700× *g*) and used for testing the biological activities. 

For ^1^H NMR profiling, 4 mg of freeze-dried extracts were solubilized in 1 mL of phosphate buffer (90 mM, pH 6.0) in D_2_O containing standard 0.01% trimethylsilylpropionic-2,2,3,3-d*_4_* acid sodium salt (TMSP). Samples were centrifuged for 10 min (1700× *g*) and 700 µL of the supernatant were transferred into NMR tubes for the analysis. 

### 3.4. Fractionation and Cynaropicrin Identification 

To better characterize predominant secondary metabolites from CycA and AuL, further purification procedures were carried out. The amount of 1 g of CycA was extracted in 50 mL of MeOH/H_2_O (1:1), sonicated for 30 min, and centrifuged for 15 min in 50 mL tubes. The supernatant was dried in a rotary evaporator, yielding 92 mg of extract, which was solubilized in 25 mL of H_2_O and subjected to liquid/liquid partition with CHCl_3_ (25 mL for three times). The organic phases were collected, anhydrificated using Na_2_SO_4_ anhydrous, filtered, and dried in the rotary evaporator. Cynaropicrin was identified by NMR experiments (^1^H NMR; HMBC, HSQC, COSY, J-res) (Appendix A and Table A1) from the CHCl_3_ fraction (yield equal to 21 mg) solubilized in CD_3_OD at a concentration of 4 mg/mL.

### 3.5. NMR Measurement and Analysis 

^1^H NMR spectra, J-resolved (J-res), ^1^H-^1^H homonuclear, and inverse detected ^1^H-^13^C correlation experiments were recorded at 25 °C on a Varian Inova 600 MHz NMR instrument (600 MHz operating at the ^1^H frequency) equipped with an indirect triple resonance probe. D_2_O was used for internal lock for ^1^H NMR profiling and CD_3_OD for the other measurements. 

For all ^1^H NMR analyses, relaxation delay was 2.0 s, observed pulse 5.80 µs, number of scans 256, acquisition time 16 min, and spectral width 9595.78 Hz (corresponding to δ 16.0). For ^1^H NMR profiling, a presaturation sequence (PRESAT) was used to suppress the residual H_2_O signal at δ 4.83 (power = −6dB, presaturation delay 2 s). 

Spectra were processed by Mestrenova software (Mestrelab Research, Santiago de Compostela, Spain), and the analysis of ^1^H NMR profiles of extracts was performed based on an in-house library and comparison with literature [55,56]. Estimation of metabolites amount in the crude extracts was calculated by comparison of diagnostic signals and TMSP (internal standard) resonating at δ 0.

### 3.6. Phenolic and Flavonoid Content 

The assays were performed in a Spectrophotometer Jasco V-530 as described by Mandrone et al. (2019) [8]. Analyses were performed in triplicate. 

### 3.7. Cell Culture and Treatment 

Human osteosarcoma cells U2OS were cultured in RPMI 1640 medium, supplemented with 10% fetal bovine serum and 2 mM L-glutamine, at 37 °C in a humidified atmosphere containing 5% CO_2_. 

Cells were plated at 1 × 10^4^ cells/cm^2^ in Petri dishes and treated with plant extracts after 24 h for all experiments.

Human umbilical vein endothelial cells (HUVEC) were plated on gelatin-coated tissue culture dishes and maintained in phenol red-free basal medium M200 (Life Technologies, Waltham, Massachusetts) containing 10% FBS and growth factors (LSGS, Life Technologies, Waltham, Massachusetts) at 37 °C with 5% CO_2_. Cells from passages 3 to 7 were actively proliferating (70%–90% confluent) when samples were harvested and analyzed. Dried extracts were dissolved in ultrapure water at 1 mg/mL and added to cell medium at the appropriate concentration and for the required time depending on the experiments.

### 3.8. MTT Assay

The cells were seeded in a 96-well plate and treated for 24 and 48 h with extracts under test in quadruplicate; controls were treated with an equal volume of water. 

A total of 20 µL of tetrazolium salt (2.5 mg/mL) in PBS was added to culture medium for 4 h, then the medium was removed and 100 µL isopropanol was added to each well. The absorbance at 570 nm of the solubilized formazan salt was determined by microplate reader (VICTOR3, PerkinElmer Life and Analytical Sciences, Milan, Italy). 

EC_50_ refers to the concentration of extract which results in a 50% reduction of cell growth, and values for different extracts were obtained by dose–response curve interpolation, using Sigma Plot 10.0 software. 

### 3.9. Cell Cycle Analysis 

DNA profiles were obtained by cytofluorimetric analysis. After 48 h of treatment, 1 × 10^6^ cells were pelleted and suspended in trisodium citrate 0.1% (*m/v*), RNase 0.1 mg/L, Igepal 0.01% (*v/v*), propidium iodide (PI) 50 mg/L. After 30 min at 37 °C in the dark, cells were analyzed by Bryte HS Biorad cytometer, equipped with Xe-Hg lamp. PI fluorescence was collected on a linear scale at 600 nm and the DNA distribution was analyzed by the Modifit 5.0 software. 

### 3.10. Analysis of p53 Expression Levels by Flow Cytometry

The cells were harvested, washed twice with PBS, fixed with 3% paraformaldehyde, washed with 0.1 M glycine in PBS, and permeabilized in 70% ice-cold ethanol. After fixing, the cells were washed with 1% BSA in PBS and incubated overnight with 1:200 anti-p53 monoclonal antibody (Upstate, MA, USA) in blocking buffer. Then, the cells were washed three times and incubated for 1 h at room temperature in 1:1000 FITC labeled secondary antibody (Sigma Aldrich, Milan, Italy). Finally, the samples were analyzed by flow cytometry. FITC green fluorescence was analyzed at 525 nm on a logarithmic scale by WinDMI 2.8 software. 

### 3.11. Caspase Activity Assay 

The enzymatic activity of caspase 3 was evaluated by using the colorimetric CaspACE assay system (Promega), according to the manufacturer’s instructions. Briefly, treated and control cells were detached, centrifuged, and resuspended in lysis buffer at 5 × 10^7^ cells/mL concentration. Cells were lysed by three freeze–thaw cycles. Cell lysates were centrifuged and supernatant fraction was collected. In a 96-well plate were added Caspase Assay Buffer (32 µL), DMSO (2 µL), DTT 100 mM (10 µL), deionized water to final volume (98 µL), and 2 µL of caspase substrate (DEVD-pNA) in each well. After 4 h of incubation at 37 °C, the absorbance at 405 nm was measured and the caspase activity was obtained. 

### 3.12. Fluorescence Microscopy 

The cells were seeded onto slides and treated with plant extracts. After 24 h of incubation, they were marked with fluorescent dye Hoechst 33,432 0.1 mg/mL for 30 min at 37 °C. Then, the samples were washed twice with PBS and fixed with 4% para-formaldehyde in PBS for 20 min at room temperature in the dark. After two washes in glycine-PBS, samples were embedded in Mowiol and analyzed by a Nikon Eclipse fluorescence microscope.

### 3.13. Statistical Analysis

Values are expressed as the mean ± SD of three independent experiments (each one performed in triplicate). Statistical analyses were performed using Graph Pad Prism 4 software (La Jolla, CA, USA). For biological assays, the statistical significance of differences among treatment groups was determined by paired Student’s T-test. For flavonoids and phenolic content, samples were compared by one-way analysis of variance (ANOVA), followed by Tukey’s honestly significant difference (HSD) post hoc test, considering significant differences at *p* values <0.05. 

## 4. Conclusions

Thirty-five extracts from Sardinian plants were screened for their antitumor potential against U2OS cells. The results showed five plants were endowed with high activity, namely: *Arbutus unedo* (AuL), *Cynara cardunculus* (CycA), *Centaurea calcitrapa* (CcA), *Smilax aspera* (SaA), and *Tanacetum audibertii* (TaA), this latter endemic to Sardinia and Corsica and still scantly investigated. The antiproliferative activity and the phytochemical profiles of these five plants were further investigated. All five extracts caused a block of cell cycle in G2/M phase, and treating with CycA and TaA, the percentage of cells in this phase was higher than 50%. The activity of TaA significantly affected p53 expression levels, while CycA activity did not involve this protein. A significant caspase activation was observed especially for CycA, followed by TaA and CcA. On the contrary, AuL and SaA did not induce apoptotic cell death, suggesting cytostatic rather than cytotoxic effects. 

Except for AuL, the other four extracts produced a marked cytotoxicity also on human endothelial vein cells, making them interesting for the additional potential to contrast angiogenesis development. On the other hand, the complete inactivity of AuL on the viability of HUVEC cells indicates its selectivity. The phytochemical analysis revealed a high presence of arbutin and flavonoids in this plant, which might be important for the measured bioactivity. Interestingly, three out of five active plants (CcA, CycA, and TaA), belong to the Asteraceae family, which is renowned for producing bioactive sesquiterpene lactones; in fact, these compounds were detected by NMR analysis. 

This study highlighted the potential use of these plants as active ingredients to develop functional food for chemoprevention or adjuvants in cancer therapy. Further phytochemical and biological studies are ongoing. 

## Figures and Tables

**Figure 1 plants-09-00026-f001:**
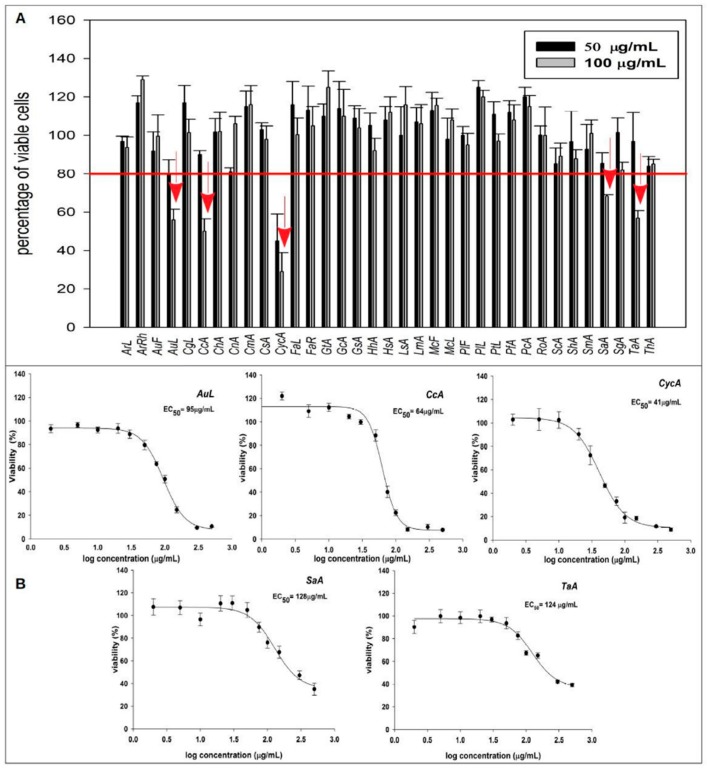
Antiproliferative effects of plant extracts on U2OS cells. (**A**) Screening of all extracts on viability of U2OS cells treated at 50 and 100 μg/mL concentrations. Bars indicate the means of six replicated experiments and represent the percentage of viable cells with respect to the control taken arbitrarily as 100%. (**B**) Dose–response curves of U2OS cell viability after treatment with the five most active extracts for 48 h.

**Figure 2 plants-09-00026-f002:**
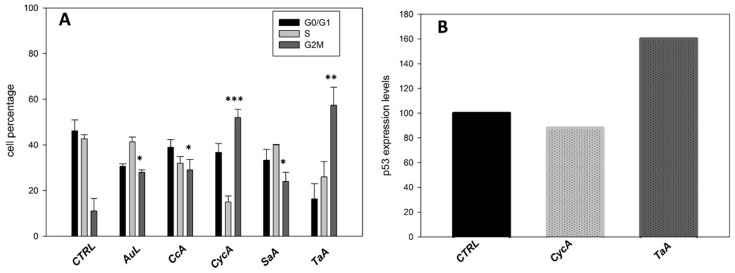
Effects of most potent plant extracts on cell cycle progression and p53 levels. (**A**) Cell cycle distribution of U2OS after 48 h of treatment with the plant extracts. Data are presented as means ± SD of three different experiments. Differences were considered significant when *p* ≤ 0.05 (* *p* ≤ 0.05; ** *p* ≤ 0.05; *** *p* ≤ 0.001). (**B**) Flow cytometric analysis of p53 protein levels after CycA or TaA treatment. The histograms represent the protein expression levels with respect to the control taken arbitrarily as 100%. Results shown are representative data from three similar experiments.

**Figure 3 plants-09-00026-f003:**
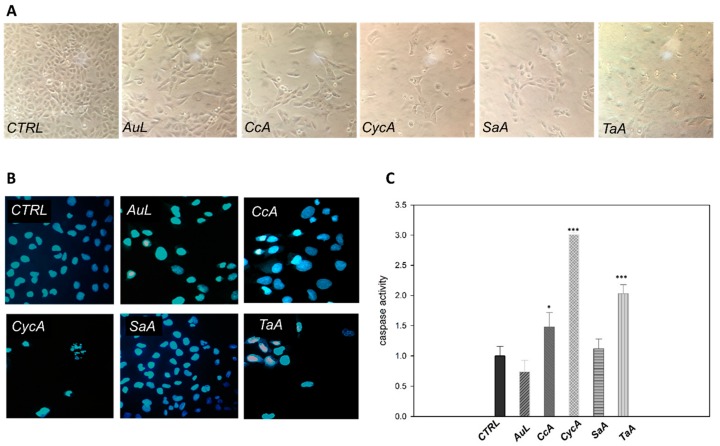
Apoptotic effects on U2OS cells treated for 48 h with the selected extracts: morphological changes and caspase activity. (**A**) Images acquired by optical microscope. (**B**) Images acquired by fluorescence microscope after HOECHST staining. (All the images were taken at the same magnification and depict microscopic fields representative of the whole cell population.) (**C**) Caspase activity assay (fold change of protein activity was calculated by taking untreated cells as a control = 1). Bar graphs represent means ± SD determined from at least three independent experiments. Differences were considered significant when *p* ≤ 0.05 (* *p* ≤ 0.05; *** *p* ≤ 0.001).

**Figure 4 plants-09-00026-f004:**
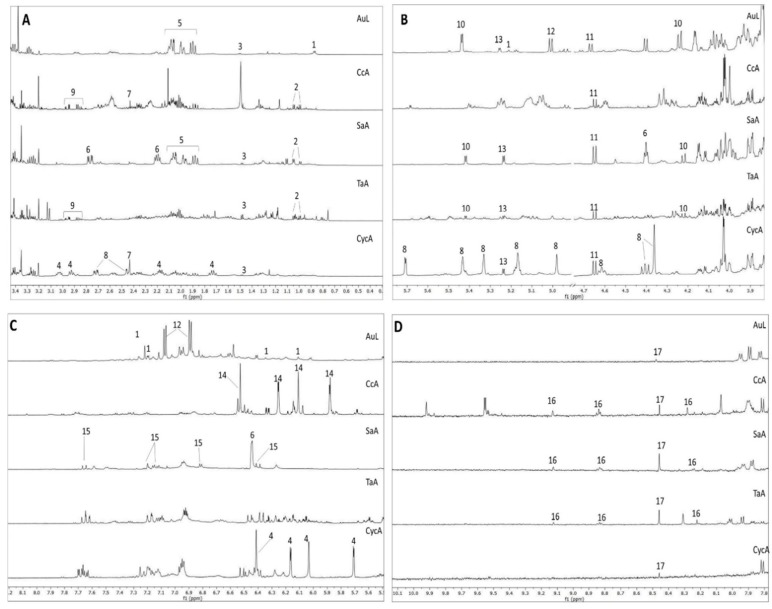
^1^H NMR extended spectral regions of AuL; CcA; SaA; TaA; CycA (from top to bottom). (**A**) region from δ 0.4 to 3.4; (**B**) from δ 3.4 to 5.4; (**C**) from δ 5.4 to 8.1; (**D**) from δ 7.8 to 10.00. Numbers indicate diagnostic signals of the most abundant metabolites: *O*-rhamnosyl flavonoid (1), valine (2), alanine (3), cynaropicrin (4), quinic acid (5), shikimic acid (6), succinic acid (7), malic acid (8), aspartic acid (9), sucrose (10), β-glucose (11), arbutin (12), α-glucose (13), sesquiterpene lactone-derivative (14), caffeic acid (15), trigonelline (16), formic acid (17). Spectra were measured in D_2_O-buffer (pH 6.0) using TMSP as standard, residual water signal has been removed.

**Table 1 plants-09-00026-t001:** Ethnobotanical use of investigated plants in Sardinian traditional medicine. Up-to-date information on plants analyzed in this work. Traditional uses related to specific organs as well as preparation are reported.

Plant Species	Plant Organ and Label	Ethnobotanical Use in Sardinia	Preparation
***Arbutus unedo* L.**	Fruits (AuF)	Astringent [22,23], blood circulation (atherosclerosis) [24]	Decoction
		wound healing [25,26]	Cataplasm
	Leaves (AuL)	Inflammations of intestine, kidney, bladder [25]	Decoction (together with fruits and roots)
		Antipyretic, diarrhea [26,27,28], intestinal pains, as a vulnerary [29], diuretic, against cystitis and nephritis [22], asthma and bronchitis [28]	Decoction
***Asphodelus ramosus* L. subsp *ramosus***	Rhizome (ArRh)	Sore throat, skin diseases [24]	Decoction
		Skin disease (chilblain) [24,29], hemorrhoids and impetigo [30]	Cataplasm
	Leaves (ArL)	Diuretic (not recommended for patients affected by rheumatisms) [22]	Decoction
***Carlina gummifera* (L.) Less.**	Leaves (CgL)	Diuretic [27,30], cholagogue, stomachic, and diaphoretic [30]	Decoction and infusion
***Centaurea calcitrapa* L.**	Aerial parts (CcA)	Antipyretic, digestive, for constipation and diarrhea [22,31]	Decoction
		Antiseptic [31]	Juice
***Centaurea horrida* Badarò ^a^**	Aerial parts (ChA)	N.D. *	-
***Centaurea napifolia* L.**	Aerial parts (CnA)	Nutritional purposes: aerial parts are often included in the diet [32]	Direct ingestion
***Cistus monspeliensis* L.**	Aerial parts (CmA)	Topically for wound healing [30]	Poultice obtained by pressing fresh leaves between two pieces of wood
***Cistus salvifolius* L.**	Aerial parts (CsA)	N.D. *	-
***Cynara cardunculus* L.**	Aerial parts (CycA)	Hepatoprotective, blood depurative, hypocholesterolemic, digestive, intestinal spasmolytic [24,33]	Decoction
		Liver diseases [28]	Infusion
***Ferula arrigonii* Bocchieri ^b^**	Leaves (FaL)	N.D. *	-
	Roots (FaR)	N.D. *	-
***Galactites tomentosa* Moench**	Aerial parts (GtA)	Nutritional purposes: aerial parts are often included in the diet [32]	Direct ingestion
***Genista corsica* (Loisel.) D ^b^**	Aerial parts, flowers (GcA)	In Corse, flowers were used as disinfectant of wounds and abrasions [32]	Infusion
***Glechoma sardoa* (Bég.) Bég. ^b^**	Aerial parts (GsA)	Treatment of respiratory diseases, chronic catarrh, bronchitis, asthma, and to heal wounds [22]	Infusion of steam and flowers in water or milk
***Hypericum hircinum* L. *ssp hircinum*^c^**	Aerial parts (HhA)	Burns and wounds healing [22]	Macerated in olive oil
		For rheumatic and sciatic pains and for dislocations and sprains [22]	Macerated in olive oil and white wine, followed by evaporation of the wine
***Hypericum scruglii* Bacch., Brullo & Salmeri ^a^**	Aerial parts (HsA)	N.D. *	-
***Lavandula stoechas* L.**	Aerial parts (LsA)	Against ringworm and skin diseases, and for wounds healing [22]	Macerated in spirit
		Treatment of migraine, vertigo, asthma, palpitation, whooping cough, laryngitis, bronchitis, rheumatism [22,23], sedative, and skin diseases [24]	Infusion
		Treatment of skin diseases [24]	Direct application of leaves
***Limonium morisianum* Arrigoni ^a^**	Aerial parts (LmA)	N.D. *	-
***Myrtus communis* L.**	Fruits (McF)	Vulnerary, cough, sedative, digestive [21]	Decoction
		Against cough and catarrh [25] and eupeptic [30]	Decoction (together with leaves)
		Digestive [25]	Macerated in spirit
	Leaves (McL)	Wound healing [30]	Dried and powdered for topical application
		Digestive and as an agent to treat respiratory ailments, as vulnerary, against hemorrhoids, to treat sweaty feet [22], catarrhal cough [26]	Infusion
		Digestive, treatment of respiratory inflammations and hemorrhoids [22]	Fresh leaves pack
		Vulnerary, cough, sedative, digestive [23], bronchitis, and asthma [28]	Decoction
***Pistacia lentiscus* L.**	Fruits (PlF)	Cutaneous inflammations [30]	Fresh-squeezed and heated for topical application
		Halitosis [22]	Fresh fruits
		Catarrhal cough, gingivitis, sore throat [30], stomachache [28]	Decoction
	Leaves (PlL)	Treatment of gingivitis, sore throat [30]	Decoction of fresh leaves to use as mouthwash
		Stomatitis, cough sedative, skin diseases [24]	Decoction
		Against ticks [25]	Fumigation
		Anticatarrhal [22], against cough and against bad breath and as an antisudorific [29]	Infusion
***Pistacia terebinthus* L. *ssp. terebinthus***	Leaves (PtL)	Catarrhal cough [26]	Decoction
***Plagius flosculosus* (L.) Alavi & Heywood ^b^**	Aerial parts (PfA)	N.D. *	-
***Ptilostermon casabonae* (L.) Greuter ^d^**	Aerial parts (PcA)	Antispasmodic [32]	Direct ingestion
***Rosmarinus officinalis* L.**	Aerial parts (RoA)	Stomachache [30], cholagogue, general tonic, against common cold, hair loss [25], inappetence, digestive, diuretic, sedative, headache, pruritus [21]	Infusion
		Hepatic [24], diarrhea [30], mucolytic, anti-inflammatory, tooth care, colic, tonic for blood pressure, joint pains [33], antitussive, antispasmodic, migraine, digestive [25], taenifuge, asthma, bronchitis; stomachic [28]	Decoction
		Antirheumatic [28]	Cataplasm
***Santolina corsica* Jord. & Fourr ^b^**	Aerial parts (ScA)	N.D. *	
***Scolymus hispanicus* L. *subsp. hispanicus***	Aerial parts (ShA)	Nutritional purposes: young stems are often included in the diet [32]	Direct ingestion
***Silybum marianum* (L.) Gaertn.**	Aerial parts (SmA)	Treatment of bleeding, diuretic, hypotensive, sudorifer in case of pneumonia and chronic catarrh [22]	Decoction (together with the seeds)
***Smilax aspera* L.**	Aerial parts (SaA)	Treatment of rheumatisms, skin diseases [24], hemorrhoids [28]	Cataplasm
		Sudorific and blood cleanser [24,27]	Decoction
		Toothache [28]	Drops of fresh-squeezed juice applied on the gums
***Stachys glutinosa* L. ^c^**	Aerial parts (SgA)	Antiseptic, antispasmodic [32]	Infusion
		Cholagogue, diuretic, and hepatoprotective [27], common cold [28]	Decoction
***Tanacetum audibertii* (Req.) DC ^b^**	Aerial parts (TaA)	Digestive, vermifuge, antiarthritic and to treat menstrual disorders [30]	Decoction
***Thymus herba barona* Loisel. ^e^**	Aerial parts (ThA)	Antitussive, expectorant, antispasmodic, collutory [22], anthelmintic, treatment of stomachache [24], sore throat, common cold, tonic and antianemic, diuretic [28]	Decoction or infusion
		Against foot perspiration and urticaria [25]	Powder obtained by crushing aerial parts
		Rheumatisms [24]	Cataplasm
		Catarrhal, antipyretic [24]	Macerated in wine
		Lung diseases [24]	Vaporization

^a^ Endemic species of Sardinia; ^b^ endemic to Sardinia and Corsica; ^c^ endemic to Sardinia, Corsica, and Tuscan Archipelago; ^d^ endemic to Sardinia (Italy), Corsica, and the Hyères islands (France); ^e^ endemic to Sardinia, Corsica, and Majorca (Spain); * N.D. = Not documented, there are no published data on the ethnobotanical use of these plants.

**Table 2 plants-09-00026-t002:** Effects of five most active plant extracts on U2OS and HUVEC cell viability. Cells were treated at 50 and 100 μg/mL concentrations for 24 h. Data are presented as means ± SD of three replicated experiments and represent the percentage of viable cells with respect to the control taken arbitrarily as 100%.

Extracts	U2OS Cells		HUVEC Cells	
	50 μg/mL	100 μg/mL	50 μg/mL	100 μg/mL
**AuL**	79.6 ± 7.7	56 ± 5.5	111.8 ± 0.63	104.7 ± 4.6
**CcA**	90 ± 2	50 ± 6.5	6.5 ± 0.29	6.4 ± 0.05
**CycA**	45 ± 1.4	29 ± 9.9	6.5 ± 0.02	7.7 ± 0.37
**SaA**	85.4 ± 5.5	68.3 ± 0.7	114.2 ± 0.08	77.3 ± 1.11
**TaA**	96.9 ± 12.9	56.9 ± 3.9	98.9 ± 0.07	50.4 ± 1.5

**Table 3 plants-09-00026-t003:** Estimated amount of metabolites by ^1^H NMR analysis.

Metabolite	Diagnostic ^1^H NMR Signal (δ) Used for the Quantification and Number of Underlying Proton/s (in Brackets)	Metabolite Quantity in the Extract (μg/mg of Extract)				
		AuL	CycA	CcA	TaA	SaA
**alanine**	1.48 (3H)	1.4	2.0	-	8.9	3.7
**arbutin**	7.07 (2H)	75	-	-	-	-
**aspartate**	2.96 (1H)	-	-	17	33	-
**caffeic acid**	7.62 (1H)	-	-	-	-	13
**cynaropicrin**	6.16 (1H)	-	159	-	-	-
**α-glucose**	5.2 (1H)	22	24	-	-	40
**β-glucose**	4.59 (1H)	36	49	-	34	88
**isoleucine**	1.06 (3H)	-	-	-	-	19
**quinic acid**	1.87 (1H)	142	43	-	-	193
**shikimic acid**	6.45 (1H)	-	-	-	-	141
**sucrose**	5.4 (1H)	138	-	-	45	79

**Table 4 plants-09-00026-t004:** List of the plants analyzed in this work. Plant name, family, considered plant organ and adopted label, harvesting date and GPS coordinates, and voucher number are reported.

Plant Name	Family	Plant Organ and Sample Label in Brackets	GPS Coordinates	HARVESTING DATE	Voucher
***Arbutus unedo* L.**	Ericaceae	Fruits (AuF)	39°45′37.8″ N 9°30′31.0″ E	December 2017	Herbarium CAG 878
		Leaves (AuL)	39°45′37.8″ N 9°30′31.0″ E	December 2017	
***Asphodelus ramosus* L. subsp. *ramosus***	Asphodelaceae	Rhizome (ArRh)	39°10′38.7″ N 9°22′50.3″ E	April 2017	Herbarium CAG 1405
		Leaves (ArL)	39°10′38.7″ N 9°22′50.3″ E	April 2017	
***Carlina gummifera* (L.) Less.**	Asteraceae	Leaves (CgL)	39°45′44.2″ N 9°40′16.9″ E	July 2018	Herbarium CAG 770
***Centaurea calcitrapa* L.**	Asteraceae	Aerial parts (CcA)	39°18′02.3″ N 8°53′39.4″ E	June 2017	Herbarium CAG 781
***Centaurea horrida* Badarò *****	Asteraceae	Aerial parts (ChA)	40°57′51.6″ N 8°12′05.0″ E	June 2017	Herbarium CAG 777
***Centaurea napifolia* L.**	Asteraceae	Aerial parts (CnA)	39°16′51.5″ N 8°56′01.5″ E	June 2017	Herbarium CAG 784
***Cistus monspeliensis* L.**	Cistaceae	Aerial parts (CmA)	39°45′44.2″ N 9°40′16.9″ E	April 2018	Herbarium CAG 135
***Cistus salviifolius* L.**	Cistaceae	Aerial parts (CsA)	39°45′44.2″ N 9°40′16.9″ E	April 2018	Herbarium CAG 135/C
***Cynara cardunculus* L.**	Asteraceae	Aerial parts (CycA)	39°18′02.3″ N 8°53′39.4″ E	April 2017	Herbarium CAG 790
***Ferula arrigonii* Bocchieri ******	Apiaceae	Leaves (FaL)	39°51′37.9″ N 8°26′05.2″ E	April 2017	Herbarium CAG 612/A
		Roots (FaR)	39°51′37.9″ N 8°26′05.2″ E	April 2017	
***Galactites tomentosa* Moench**	Asteraceae	Aerial parts (GtA)	39°46′16.7″ N 9°30′41.6″ E	September 2018	Herbarium CAG 789
***Genista corsica* (Loisel.) DC ******	Fabaceae	Aerial parts (GcA)	39°49′35.0″ N 9°20′27.5″ E	May 2017	Herbarium CAG 286
***Glechoma sardoa* (Bég.) Bég ******	Lamiaceae	Aerial parts (GsA)	39°57′33.5″ N 9°19′13.3″ E	June 2017	Herbarium CAG 1104
***Hypericum hircinum* L. *ssp hircinum^+^***	Hypericaceae	Aerial parts (HhA)	39°46′55.8″ N 9°30′52.5″ E	June 2018	Herbarium CAG 232
***Hypericum scruglii* Bacch., Brullo & Salmeri *****	Hypericaceae	Aerial parts (HsA)	39°45′57.4″ N 9°30′41.8″ E	June 2018	Herbarium CAG 239/C
***Lavandula stoechas* L.**	Lamiaceae	Aerial parts (LsA)	39°45′44.2″ N 9°40′16.9″ E	April 2017	Herbarium CAG 1067
***Limonium morisianum* Arrigoni *****	Plumbaginaceae	Aerial parts (LmA)	39°54′33.3″ N 9°24′41.0″ E	December 2017	Herbarium CAG 909/G
***Myrtus communis* L.**	Myrtaceae	Fruits (McF)	39°45′44.2″ N 9°40′16.9″ E	December 2018	Herbarium CAG 514
		Leaves (McL)	39°08′22.2″ N 8°58′08.9″ E	April 2018	
***Pistacia lentiscus* L.**	Anacardiaceae	Fruits (PlF)	39°45′44.2″ N 9°40′16.9″ E	December 2017	Herbarium CAG 280
		Leaves (PlL)		December 2017	
***Pistacia terebinthus* L. *ssp. terebinthus***	Anacardiaceae	Leaves (PtL)	39°47′38.8″ N 9°30′38.3″ E	June 2018	Herbarium CAG 279
***Plagius flosculosus (* L.) Alavi & Heywood ******	Asteraceae	Aerial parts (PfA)	39°21′45.2″ N 8°32′24.1″ E	July 2017	Herbarium CAG 743
***Ptilostemon casabonae* (L.) Greuter *^++^***	Asteraceae	Aerial parts (PcA)	39°53′52.7″ N 9°26′31.8″ E	June 2018	Herbarium CAG 796
***Rosmarinus officinalis* L.**	Lamiaceae	Aerial parts (RoA)	40°34′10.1″ N 8°22′57.0″ E	May 2017	Herbarium CAG 1091
***Santolina corsica* Jord. & Fourr******	Asteraceae	Aerial parts (ScA)	40°32′30.6″ N 9°36′09.4″ E	November 2017	Herbarium CAG 732/A
***Scolymus hispanicus* L. *subsp. hispanicus***	Asteraceae	Aerial parts (ShA)	39°03′25.9″ N 8°58′46.3″ E	June 2018	Herbarium CAG 812
***Silybum marianum* (L.) Gaertn.**	Asteraceae	Aerial parts (SmA)	39°16′51.5″ N 8°56′01.5″ E	May 2017	Herbarium CAG 801
***Smilax aspera* L.**	Smilacaceae	Aerial parts (SaA)	39°10′38.7″ N 9°22′50.3″ E	May 2017	Herbarium CAG 1414
***Stachys glutinosa* L*.^+^***	Lamiaceae	Aerial parts (SgA)	39°55′46.1″ N 9°27′10.7″ E	June 2017	Herbarium CAG 1099
***Tanacetum audibertii* (Req.) DC******	Asteraceae	Aerial parts (TaA)	40°02′07.9″ N 9°17′59.1″ E	August 2018	Herbarium CAG 737/A
***Thymus herba barona* Loisel ^§^**	Lamiaceae	Aerial parts (ThA)	39°56′01.2″ N 9°19′56.9″ E	June 2017	Herbarium CAG 1065

* Endemic species of Sardinia; ** Endemic to Sardinia and Corsica; ^+^ Endemic to Sardinia, Corsica, and Tuscan Archipelag; ^++^ Endemic to Sardinia (Italy), Corsica, and the Hyères islands (France); ^§^ Endemic to Sardinia, Corsica, and Balearic Islands (Spain).

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
