# Peer review of "Antitumor Potential and Phytochemical Profile of Plants from Sardinia (Italy), a Hotspot for Biodiversity in the Mediterranean Basin"

_plants, 2019, doi:10.3390/plants9010026_

Round 1

Reviewer 1 Report

The manuscript is interesting, but I found several problems, mainly in the treatment of data (the statistical analyses have not been performed except for antioxidant activity and the number of experiments is not reported). Therefore, in my opinion, the paper could be published on Molecules after several mandatory revisions.

My comments are:

Lines 34-65:
The introduction must be improved. It is too generic and the cited articles are often published more than 5 years ago. Therefore, in some cases, they do not represent the state-of-art. Moreover, in my opinion, the authors have to underline the difference between the studies about extracts and those about isolated secondary metabolites from plants and/or microorganisms. Several exemples have been published very recently about cardiovascular, Alzheimer's diseases and cancer therapies.

Line 69:

Figures and Tables in SI file must be ordered as presented in the main manuscript. I find Table S1 useful for readers. Therefore, I suggest to shift it in the main text.

Line 72:

Only in Sardinia? If there is literature for other uses of these plants worldwide, it should be mentioned. If not, in my opinion, the word "Sardinian" should be erased.

Line 82:

"Data not shown" are perfect to be included, this time, in SI file.

Line 105:

No statistics are included. The authors should perform these analyses and comment the results according the not significative/significative differences. Without statistics, every speculation is not supported!

Line 1-25:

The graphics are clear and the speculations seem correct. But... not statistic analysis again! And what about the number of experiment performed? (SD is reported!)

Line 146:

Table S2 should be included in the main manuscript, even if a figure with histograms would be better (with statistic analysis, of course!)

Figure S3:

Please correct Cna/Can

Lines 249-251:

I do not understand. This sentence, in my opinion, has low "scientific soundness". The authors should report the number of experiments and the statistic analyses (in every part of the text!)

Line 284:

What about the purity of these compounds? Did the authors perform HRMS or Elementar Analysis? In my opinion this should be mandatory.

Line 380:

Statistic analyses are mandatory for every speculation.

Lines 389-390:

The antioxidant activity can be useful also for therapies of other pathologies (such as cardiovascular and Alzheimer's diseases). I suggest to report it for the future perspectives of the work.

Author Response

Referee1:

Thanks very much for your comments, which helped to improve the overall quality of our work.

Please find below our point-by-point responses to the queries.

Lines 34-65:

The introduction must be improved. It is too generic and the cited articles are often published more than 5 years ago. Therefore, in some cases, they do not represent the state-of-art. Moreover, in my opinion, the authors have to underline the difference between the studies about extracts and those about isolated secondary metabolites from plants and/or microorganisms. Several exemples have been published very recently about cardiovascular, Alzheimer's diseases and cancer therapies.

Reference 15 in the introduction (which was dated 2005) has been replaced with a more recent review on natural products and cancer treatment (dated 2019). We have inserted lines 59-61 where we mentioned the potential of crude plants extracts in combatting cancer, supported by references 16, 17, both dated 2019.  

Line 69:

Figures and Tables in SI file must be ordered as presented in the main manuscript. I find Table S1 useful for readers. Therefore, I suggest to shift it in the main text.

Supporting material has been ordered as recommended, inserting in the text Table S1, which is now Table 1.

Line 72:

Only in Sardinia? If there is literature for other uses of these plants worldwide, it should be mentioned. If not, in my opinion, the word "Sardinian" should be erased.

Yes, the literature refers only to Sardinia, because the plants mentioned in these lines are endemic and exclusive of this Island. Thus, as suggested, we removed the word "Sardinian", which could generate misunderstandings.

Line 82:

"Data not shown" are perfect to be included, this time, in SI file.

These data have been inserted in a table in SI (now Table S1).

Line 105:

No statistics are included. The authors should perform these analyses and comment the results according the not significative/significative differences. Without statistics, every speculation is not supported!

Statistics has been performed and discussed in text and figures. A specific paragraph on statistical analysis has also been inserted.

Line 1-25:

The graphics are clear and the speculations seem correct. But... not statistic analysis again! And what about the number of experiment performed? (SD is reported!)

Statistics has been performed and number of experiments mentioned.

Line 146:

Table S2 should be included in the main manuscript, even if a figure with histograms would be better (with statistic analysis, of course!).

We included Table S2 in the text (now Table 2). In our opinion, adding one more graph/ figure in the text (which is already very long right now) would have weighed down the manuscript. Anyway, if the Editor prefers to transform Table 2 in a Graph, we have already prepared it, and we are ready to quickly make this change.

Figure S3:

Please correct Cna/Can

Done

Lines 249-251:

I do not understand. This sentence, in my opinion, has low "scientific soundness". The authors should report the number of experiments and the statistic analyses (in every part of the text!)

We removed this unclear sentence and we have mentioned number of experiments in the caption of each figure, along with all the useful information to understand how statistics was performed.

Line 284:

What about the purity of these compounds? Did the authors perform HRMS or Elementar Analysis? In my opinion this should be mandatory.

Our goal was not to isolate pure compounds, and actually, we didn’t. The pre-purification procedures adopted were performed only to confirm the structure of the most prominent metabolites in the extracts, which were than quantified by q-NMR. Since the detected metabolites are not new molecules, as well as they were previously identified in these plants, we don’t feel HRMS is required, 1D and 2D NMR analysis should be sufficient in this case.

Line 380:

Statistic analyses are mandatory for every speculation.

Done

Lines 389-390:

The antioxidant activity can be useful also for therapies of other pathologies (such as cardiovascular and Alzheimer's diseases). I suggest to report it for the future perspectives of the work.

We agree that antioxidant activity is an important additional information for bioactive plant extracts. However, this manuscript is already quite long, with several biological activities measured and with 56 bibliographical references. Moreover, the phenolic and flavonoid content (which has been determined) is usually a good indicator of the expected in vitro antioxidant activity. For this reason, unless it will be considered mandatory by the Editor we decided not to perform more experiments on these extracts. However, our work on Sardinian plants is ongoing and your is a good suggestion for a new project.

Reviewer 2 Report

The article “Antitumor potential and phytochemical profile of plants from Sardinia (Italy), a hotspot for biodiversity in Mediterranean basin” of Cappadone et al. shows the cytotoxic activity of 35 extracts derived from Sardinian plants, some of them poorly studied endemisms.

The text is well organized, and the style is fine, although some minor spell check is required. Some typographical mistakes are highlighted in the attached document. Please, use the international system of units abbreviations.

Some points should be clarified to improve the manuscript:

The standard deviation is lacking in figure 2. Statistical analysis should be performed in order to determine the statistically significant differences between the treated and the control groups.

Author Response

Referee 2

Thanks for you comments and revision, which helped us to improve our work.

Please find below our reply point by point to your comments.

The text is well organized, and the style is fine, although some minor spell check is required. Some typographical mistakes are highlighted in the attached document. Please, use the international system of units abbreviations.

We have modified what the referee highlighted.

Some points should be clarified to improve the manuscript:

The standard deviation is lacking in figure 2. Statistical analysis should be performed in order to determine the statistically significant differences between the treated and the control groups.

Statistics has been reported in Fig. 2A and others and discussed in the text and figures captions. In Fig. 2B, been the histograms representative of percentages, inserting error bars would not be appropriate.  

Reviewer 3 Report

Manuscript ID

plants-667488

The manuscript entitled „Antitumor potential and phytochemical profile of plants from Sardinia (Italy), a hotspot for biodiversity in Mediterranean basin” is interesting and suitable for the publication in Plants

However, I have a few comments/questions:

Figure 1A: the Y-axis is described as percentage of viable cells, but the scale is not in percentage.
Of course, the results are clear and the intentions of the authors also, but this is an obvious mistake

line 121, Fig 3C, lines 361-368:

which caspase activity was determined? - it does not result from the description,
based on the name of the substrate, it is caspase 3
provide the correct name of the test used for analyzes - CaspACE is probably not correct

Please explain:

Casp-3 has been shown to be the major executive caspase of most apoptosis pathways, and an increase in its activity was considered a marker of the progression of apoptotic death.
Caspase activity increased in cells treated especially with CycA, while no activation of p53 was found in them.
What other mechanism for caspase activation can be considered? Please include this explanation in the manuscript.

I understand that the research is innovative and there are not many results regarding the antitumor activity of the analyzed plants. However, this does not explain the lack of discussion of the obtained results. Maybe you should compare your own results with the research of other scientists, using the same cell line and other plant extracts or chemotherapeutics

line 29 – O-o-rhamnosyl should be, I suppose

Author Response

Referee 3

Thanks very much for your comments, which helped to improve the overall quality of our work.

Please find below our point-by-point responses to the queries.

Figure 1A: the Y-axis is described as percentage of viable cells, but the scale is not in percentage.

Of course, the results are clear and the intentions of the authors also, but this is an obvious mistake.

Thanks for your observation. We have corrected it.

line 121, Fig 3C, lines 361-368:

which caspase activity was determined? - it does not result from the description, based on the name of the substrate, it is caspase 3 provide the correct name of the test used for analyzes - CaspACE is probably not correct

We have specified that caspase 3 activity was determined. The name of the kit used for analysis is exactly: colorimetric CaspACE assay system (Promega). 

Please explain:

Casp-3 has been shown to be the major executive caspase of most apoptosis pathways, and an increase in its activity was considered a marker of the progression of apoptotic death.

Caspase activity increased in cells treated especially with CycA, while no activation of p53 was found in them.

What other mechanism for caspase activation can be considered? Please include this explanation in the manuscript.

Done in lines 118-122.

I understand that the research is innovative and there are not many results regarding the antitumor activity of the analyzed plants. However, this does not explain the lack of discussion of the obtained results. Maybe you should compare your own results with the research of other scientists, using the same cell line and other plant extracts or chemotherapeutics

We already discussed the bioactivity of the five selected plants, and for two of them there is no literature available, being rare plants.  This work was based on a relatively high number of plants, we understand how important is to provide a broad spectrum discussion, however the work, in the present form, is already quite long (with 56 bibliographic references) and we are afraid that an increase in the discussion and bibliographic notes is going to make the overall manuscript too heavy. Moreover, comparing row extracts activities with pure drugs, like chemotherapeutics, is far from the objective of this work. Our studies on Sardinian plants are ongoing and we wish to be able to give continuity to this data in the future, focusing on a single plant and its active principles, in that case, we will of course focus the discussion also on the comparison with specific synthetic drugs and/or other natural products.

line 29 – O-o-rhamnosyl should be, I suppose

Here we missed ‘y’ in the world ‘rhamnosyl’ and we have corrected it. The ‘O-’ must remain italic capital letter, for it indicates the O-glycosidic bond in the molecule. Lower case letter (-o-) would indicate an orto substitution on an aromatic ring, and this is not the case.

Reviewer 4 Report

see attachment

Author Response

This response was previously sent only by email. Please find it attached as word file.

Round 2

Reviewer 4 Report

Such optimistic, unpredictable and unique research results presented by the authors may be the result of the presence in the plant and isolation of the unique sugar derivative of the commonly known compound like Quercitrin.

Unfortunately, scientific soundness is poor solely because of the non-classical (non-professional) description of some NMR data. We describe all protons based on an analytical sample. The authors believe in the correctness of their style, but may have a crude sample or a pure derivative of the described molecule of the Quercitrin. functionalized on the sugar hydroxy groups by sugared motif.

Generally, I receive the publication positively and recommend it for publication in an international journal in Plants after removing minor errors and strict compliance with the general style of NMR data presentation at the body of the work. Like this text, further comments below are presented in the same color.

We would like to thank the reviewer for the careful checking, which helped to improve the quality of the manuscript.

Please find below or response point-by-point to the reviewer queries.

Authors still incorrectly use a short pause ( - ) and correctly average pause  ( – ) between numbers at references.

The work (plants-667488) entitled “ Antitumor potential and phytochemical profile of plants from Sardinia (Italy), a hotspot for biodiversity in Mediterranean basin” submitted for review by Dr Manuela Mandrone and co-workers addressed to Plants magazine seems to relate to a momentous topic that is the subject of research in modern laboratories, and its content can be published in the journal "plants", after removing a number of serious errors and ambiguities. Despite the good intentions of the authors, according to the nature of the 1H NMR analytical technique used, the method is not strictly quantitative but allows only to indicate the content of metabolites with an accuracy much lower than that given in Table 1. At least some decimal places can be removed.

We are not sure that we fully understood what the referee means with ‘some decimals’, in Table 1 (now Table 3) values have only one decimal. Although we agree that 1H NMR has to be considered more semi-quantitative than quantitativetechnique, we feel that one decimal is proper and the value doesn’t need to be further approximated.

In general, the values given are too accurate. They are only the result of mathematical calculation.

For example, correctly two significant numbers 1.4, 2.0, 8.9, and 3.7 respectively are given for alanine. Also the value 28 with the correct accuracy for quercitrin was given. Unfortunately, all other assets were recorded with three or four significant places, for example 75.2 (for arbutin), and 159.4 (for cynaropicrin), respectively, correctly, according to calculations. In all these cases, please remove the decimal digits, i.e. those after the decimal point (please round off).

.

Another disadvantage is treating the unusually large table 2 as an integral part of the work, and the information about this table and the discussion is contained only in a few lines. Please analyze the content of the table 2 more detailed.

In our opinion, Table 2. (now Table 4.), although being quite long, it is integral and essential part of the main text, belonging to methods and material of this work. We believe that this table is important for the readers to have easy access to vouchers and full names of all the plant tested in this work. Information present in this table are not discussed because it belongs to the material section, therefore it is not an element of discussion.

Ok.

Another significant disadvantage is the addition of a more important table S1 with references in the additional materials (SI), which should be fully cited in the main part of the work with full discussion. In addition, information in this table that is not documented requires the indication of a specific source of information in each isolated case, or removal of information from the scientific work sent for consideration in the journal "Plants" published by the renowned publisher "MDPI".

As suggested also by other reviewers Table S1 has been inserted in the main text as Table 1. All the data we provided in this Table were carefully cheeked before we submitted our manuscript to Plants. The information we reported in ex Table S1 (now Table 1) were and are fully documented by exhaustive bibliographical references. Two of the authors of this manuscript, namely Prof. F.P. and Dr. C.S. are experts of Sardinian ethnobotany, and thanks to their contribution, we are absolutely confident in the reported data about traditional uses of Sardinian plants. When in the table was/is written ‘not documented’ (regarding the ethnobotanical use of a specific plant), it is because they are rare species and there is no scientific literature (nor surveys) available on the ethnobotanical uses of these plants. However we have better clarified this point in Table footnotes.

Ok.

Unfortunately, the lack of documentation of significant structural evidence is serious mistake. With so many experimenters, this leads to mistakes diminishing the rank of work. Please add 2D maps, such as HMBC and COSY to SI, and in the main part of the publication please add figure containing the structures of all these more complex natural compounds together with the substituent numbering system.

The molecules reported in the main text are very common and renowned, and being the text already quite long  we don’t really feel to insert one more figure only for the structure of these simple metabolites.

These molecules are common in plants, but this interesting publication is addressed to a wide audience for a deeper reading, and the nomenclature of natural compounds is not obvious to everyone. Please disclose numbering system at the body of the work, especially for sugar derivatives.

In addition, in the main part of the work, both key isolated molecules, quercitrine and myricitrin, must have complete data and reference to the source literature, because both are not new.

Bibliographic references for these renowned compounds have been inserted in the text (Biblio 55 and 56).

But the data are still incomplete - it should be noted why and it is best to transfer the description to additional materials.

Most importantly, chemical shifts and spin-spin coupling constants are physicochemical constants. Lack of correlation the data at Table S3 and Appendix A undermines the thesis of the work. A suggest of a thorough revision of the data before sending it again to the Plants.

We have revised Table S3 (now Table S2), we detected the mistakes, which unfortunately were present. We decided to give revised coupling constant values of cynaropicrin only in Appendix A , to avoid data repetition.

Please harmonize the data.

See also more detailed comments below if other opinions were flattering, or the authors would plan a thorough revision: At line 80 is “(SaA) Tanacetum“ but there is no comma “ , “.

Corrected

Ok.

At line 81 is: … To calculate the EC50 concentration of the selected plants, … . Please define the "IC50" once.

In our work we don’t deal with ‘IC50’ but only ‘EC50’, we have now defined it in Material and Methods chapter at line 357.

Ok.

At line 83 is: … (Fig. 1B). … . It's OK, but the dot ( . ) and space ( ) between Fig and 1B are bolded incorrectly.

Corrected

Ok.

At lines 83–84 is: … treating the cells for 48 h with EC50 concentration. ….

1) What exactly were the cells treated with?

2) It is not clear why “with EC50”, please explain the choice to the reader best by referring to the state

of knowledge.

1) The cells were treated with the extracts at EC50  concentration. We have better specified this concept in the text (line 96)

2) Once established the EC50concertation, the following experiments were addressed to assess the modification on cell cycle and the caspases activation eventually induced by the treatment. In order to do this, it is common praxis to use the EC50concentration, we don’t feel this is something we have to further explain in the main text.

For item 1, it's good.

For sub-item 2, the authors' decision is acceptable.

At line 91 two dot characters are incorrectly inserted at the end of the sentence.

Please remove one of the dots.

Done

Ok.

At line 273 is “by 1D and 2D NMR experiments”. According to my knowledge, 1D NMR generally is spectrum for example, excluding 1D dept experiments. Please specify.

We have removed ‘1D and 2D’ and left only ‘by NMR experiments.’

Ok.

At line 285 is Quercitrin complete 1H NMR spectrum in numerical form, at least, the authors pledge, for example in additional materials in footnote c under table S3, but is lack of 3’’ and 5’’ protons signals. In 13C NMR spectrum I also did not see a set of signals like 4’’ and 5’’.

We are not sure we got the point because of the twisted form of this comment. However,the protons 3’’ -5’’  and 4’’ and 5’’, being part of the sugar ring are in a very crowded spectral region, that’s why we have not reported their values. A number of published NMR-based metabolomic studies identify simple compounds like these flavonoids even in the complex mixture without providing all the values of the sugar ring protons. We did some step of purification and we obtained a fraction containing mainly these two flavonoids and we did 2D NMR experiments to confirm what was already evident from 1H NMR, for instance to be sure they were not glucuronides.  They are very simple and almost ubiquitous  molecules in plants, thus for us the provided data should be enough for these kind of molecules.

Please present of this information in a clear way, in the immediate vicinity of the NMR data presented at the Supporting Information only, because at body of the publication usually gives the characteristics of the molecule as a chemical individual, i.e. for an analytical sample.

At line 303 is “power=”, but there is no space character immediately before the equal sign ( = ).

Space inserted

Ok.

At lines 404–406 in the Appendix A the presented NMR data of cynaropicrin does not critically correlate with data from Table S2. Chemical shifts and spin-spin coupling constants are physicochemical constants. Lack of correlation undermines the thesis of the work until it is rejected, including a suggestion of a thorough revision of the data before sending it again. For example chemical shifts of the protons number 15a and 15b given as 5.43, 5.33 and 6.30, 5.96 respectively. Similarly for spin-spin coupling constants are given J = 4.91 Hz and J = 1.4 Hz respectively. Which is beyond the allowable acceptable measurement error and strongly suggests a serious error. In addition, the method of recording data is random. For position 1 for 1H the order is incorrect and no symbol “Hz” at the end. The authors place a space, or not, after an equal sign “=” by accident.

As reported above we are sorry for our mistake, we have revised Appendix A.

Appendix A at line404–406 – specific comments are below:

In position 1 is: … J=11.53; 6.84; 10.65 Hz) … , but it should be in order from the largest number to the smallest number: … J=11.53; 10.65; 6.84 Hz) … 

In position 2 is: … J=13.14; 6.84; 7.50; Hz) … , but it should be in order from the largest number to the smallest number and one semicolon ( ; ) (after the number before the space character) is to be removed: … J=13.14; 7.50; 6.84 Hz) …   

In both positions 14a and 14b is:  … J= 2.09 … , but should be: … J=2.09 … . The space between the equal sign ( = ) and the number 2.09 can be removed to preserve the style as in most cases.

In both positions 15a and 15b is:  … J= 1.97 … , but should be: … J=1.97 … . The space between the equal sign ( = ) and the number 1.97 can be removed to preserve the style as in most cases.

In position 2b is: … 8.80; Hz) … , but should be: … 8.80 Hz) … . The semicolon (;) (after the number before the space character) should be removed.

I don't change the style above, but only the lack of compliance with the authors' style.

At line 414, position 3, a double dot is inserted incorrectly before the title.

Done

Ok.

At line 442 is “A Review”. Should the word "review" be capitalized?

Corrected

Is not corrected!

At line 464 is : … A Review … cancer, … , should be: … A review … cancer. … .

The word review should be in lowercase, moreover there should be a period ( . ) at the end of the publication title, not a comma ( , ).

At line 447 in position 13 is ” H.; 447 Combination”. In other cases, the last initial was completed only

by a period.

Corrected

Ok.

Line 457. Should there be a comma after the word Cancer?

Corrected

Ok.

Lines 461, 497, 517. Usually the volume/s are in italics.

Corrected

Ok.

Line 465. There should be a medium pause mark ( – ) between the page numbers.

Corrected

Not corrected! The authors still incorrectly use a short pause ( - ) and correctly average pause ( – ) between numbers.

At line 505 is: … 57-164. …, but should be: … 57–164. …,

Similarly at lines: 497, 481, 502, 510, 518, 520, 524, 536, and 588. Please corrected.

Similarly, put a mark “ – “ in the table A1 between the numbers and for AuL is 85-150 but it should be (85–150) in brackets.

Line 474. Too many first uppercase letters in the title.

Corrected

Ok.

Line 476. The second part of the name is usually also written in capital letters.

Corrected

Ok.

Line 488. Please check how the authors' names are written.

Corrected

Ok.

Fig. S1 is no space between Fig. and S1 and is no spaces before A and B.

Corrected

At SI at figure S1 are: … Fig.S1. Comparison …, should be: … Fig. S1. Comparison … .

At SI at figure S1 are: … (AuL).A) …, should be: … (AuL). A) … . There is no space before point A - this may be the fault of the data processing system.

At SI at figure S1 are: … spectrum,B) …, should be: … spectrum, B) … . There is no space before point B - this may be the fault of the data processing system.

Table S2 is no space between S2 and the title.

Corrected

Unfortunately, still is no space between “S1.” and “Spectral”.

Table S3 is no space between S3 and the title, please define “ov” in the footnote “a” is a reference to

the missing table 6.

‘Ov’ is commonly used abbreviation in NMR metabolomics works, it stands for “overlapping” signals, being so commons we feel not to insert one more footnote, which will weight down the overall table. Table 6, previously mentioned in footnotes of this table (now Tab. S2), was a mistake, which has been corrected. It referred to Appendix A.

Ok.

In the last footnote to literature in SI, the range of pages is to be revised.

Corrected.

Ok.

Also:

In line 73 is: “Uses in Sardinian traditional medicine.”

Incomplete sentence, what was used?

At table 1 some sentences end with a dot, while others don't. Please unify.

At line 162 is: “Fig.1,”, but should be: “Fig. 1,”. Pleas add space character.

At table 2 generally is no space characters before and after the mathematical character “±”, but in the first case both spaces are given. Please unify.

Additional materials:

At additional materials is no title page. Please complete with a list of authors, affiliations and a list of contents.

Is “Table S.1.” but should be “Table S1.” Delete unnecessary dot.

Is ”assay.Data” but should be ”assay. Data”. Add the missing space.

At Table S2 is “1.20 (1H-2”. The chemical shift given for position 2 in the benzene ring does not make sense. A chemical shift at 7.20 ppm may occur in the benzene ring (rich in electrons).

In table 2 the data is a serious mess. Please unify the style.

After the spin-spin coupling constant a hertz mark (Hz) is always given and always put a space before and after the equal sign ( = ).  The next data, given in brackets, are separated by a comma. We insert a short pause ( - ) after the number of protons.  Enter "J =" before the spin-spin coupling constant value, see quercitrin

At Fig. S3. Is “spectra”, but should be “experiments”.

At Fig. S2. no spaces after A) . Please enter a missing space. Similarly at Fig. S1.

Is: “Fig.S3.COSY”, but should be “Fig. S3. COSY”. Insert both missed spaces

Lines 115 and 118 correctly entered … (Fig.3A) … and … (Fig.3B) … respectively, however, there are no spaces between … (Fig. and 3A) and between (Fig. and 3B), respectively.

At line 154 is “(Fig. 4).” The parenthesis close sign is not correctly bolded.

Done

Ok.

At lines 158–160 is … Quantitative 1H NMR analysis. …, but should be like … Estimating the amount of metabolites by 1H NMR analysis. … or otherwise at the suggestions of the authors. Unfortunately, despite the good intentions of the authors, according to the nature of the analytical technique used, even in the analysis of pure substances, the integration error often appears in the second significant place (and for protons that cannot be exchanged during the measurement), and with a rich mixture, foreign signals also interfere with the measurements of the integration value. The 1H NMR technique does not allow the accuracy of the results given in Table 1 for selected metabolites. The decimal places are necessarily to be removed, except for alanine, for which two significant places are correctly given.

We replaced it with ‘Estimated amount of metabolites by 1H NMR analysis.’ We keep not getting the overall point of the decimals. For alanine, as for all the other metabolites reported in the Table 3,values have only one decimal. We believe that adopt further approximation is not needed.

At line 192 is: Table 3. Estimated amount of metabolites by 1H NMR analysis. Results were obtained by comparison of metabolites diagnostic signals and TMSP (internal standards) for the five active extracts.

Shouldn't be: … Table 3. Estimated amount of metabolites by 1H NMR analysis. Results were obtained by comparison of metabolites diagnostic signals calibrated on TMSP (internal standard) for the five active extracts. … , or … shouldn’t be: Table 3. Estimated amount of metabolites by 1H NMR analysis. Results were obtained by comparison of metabolites diagnostic signals for the five active extracts. … .

Information “calibrated on TMSP (internal standard)” should be found in the subsection 3. “Materials and methods”

Spelled correctly “ For alanine, as for all the other metabolites reported in the Table 3,values have only one decimal. We believe that adopt further approximation is not needed.”

The number of significant places is not related to the number of decimal places. Significant places start with the first digit (other than zero). After the decimal point we have decimal places (also zero). Due to the accuracy of measurements, maybe please correct the numbers of metabolite quantity in the extract at table 1. in order: 1.4, 2.0, 8.9, 3.7, 75, 17, 33, 13, 159, 22, 24, 40, 36, 49, 34, 88, 28, 142, 43, 193, 141, etc.

At lines 187–192 the NMR data are discussed, but there is no evidence. Please add 2D maps, such HMBC and COSY, to SI and in the main part of the publication a figure containing the structures of all the more complex natural compounds discussed together with the substituent numbering system.

To better follow our discussion on the (still unknown) compound from CcA, we have reported the 2D map (Fig. S3-S4). However, the identified molecules reported in the main text are very common and renowned, and being the main text already quite long we don’t really feel to insert one more figure only to repost the structure of these metabolites. In SI we already give also Fig. S2 reporting 1H NMR of cynaropicrin, with its structure and complete assignation of all protons, despite this is known to be the major compound ofCynaracardunculus(biblio 45) and its structure and NMR data are reported in a number of published papers.

The requested graphic data was placed, which allowed to check the correctness - it's okay.

At line 191 is “whose proton resonates at δ 2.50,”. As far as the discussion given is about the guaianetypesesquiterpene lactone cynaropicrin (as inform at line 172), as I presume, in Fig. S2 A and C there is no signal at 2.5 ppm.

Fig. S2 refers to cynaropicrin (from CycA), while in these lines we are discussing another (not fully elucidated) compound from CcA. We now provided the 2D spectra to better follow our discussion in SI (fig. S3-S4).

Ok, but at the line 191 is mistakenly the dot mark at the bold style.

Author Response

Please find attached, as a word file, our response to the reviewer 4 round 2.
